# Fabrication and Performance of Phase Change Thermoregulated Fiber from Bicomponent Melt Spinning

**DOI:** 10.3390/polym14091895

**Published:** 2022-05-06

**Authors:** Zenan Liu, Diefei Hu, Juming Yao, Yan Wang, Guoqing Zhang, Dana Křemenáková, Jiri Militky, Jakub Wiener, Li Li, Guocheng Zhu

**Affiliations:** 1College of Textile Science and Engineering, Zhejiang Sci-Tech University, Hangzhou 310018, China; liuzenan26@163.com (Z.L.); hudiefei1027@163.com (D.H.); amywang1021@hotmail.com (Y.W.); 2College of Materials Science and Engineering, Zhejiang Sci-Tech University, Hangzhou 310018, China; yaoj@zstu.edu.cn (J.Y.); zgq@zstu.edu.cn (G.Z.); 3Zhejiang-Czech Joint Laboratory of Advanced Fiber Materials, Zhejiang Sci-Tech University; Hangzhou 310018, China; 4College of Materials Science and Chemical Engineering, Ningbo University, Ningbo 315201, China; 5Faculty of Textile, Technical University of Liberec, 46117 Liberec, Czech Republic; dana.kremenakova@tul.cz (D.K.); jiri.militky@tul.cz (J.M.); jakub.wiener@tul.cz (J.W.); 6Institute of Textiles and Clothing, The Hong Kong Polytechnic University, Hong Kong 999077, China; li.lilly@polyu.edu.hk

**Keywords:** phase change material, thermoregulated fiber, melt spinning, enthalpy

## Abstract

High thermostability of phase change materials is the critical factor for producing phase change thermoregulated fiber (PCTF) by melt spinning. To achieve the production of PCTF from melt spinning, a composite phase change material with high thermostability was developed, and a sheath-core structure of PCTF was also developed from bicomponent melt spinning. The sheath layer was polyamide 6, and the core layer was made from a composite of polyethylene and paraffin. The PCTF was characterized by scanning electron microscopy (SEM), thermal analysis (TG), Fourier Transform Infra-Red (FTIR), X-ray diffraction (XRD), differential scanning calorimetry (DSC) and fiber strength tester. The results showed that the core material had a very high thermostability at a volatilization temperature of 235 °C, the PCTF had an endothermic and exothermic process in the temperature range of 20–30 °C, and the maximum latent heat of the PCTF reached 20.11 J/g. The tenacity of the PCTF gradually decreased and then reached a stable state with the increase of temperature from −25 °C to 80 °C. The PCTF had a tenacity of 343.59 MPa at 0 °C, and of 254.63 MPa at 25 °C, which fully meets the application requirements of fiber in textiles.

## 1. Introduction

Phase change materials (PCMs) can store and release thermal energy by changing their form. Phase change thermoregulated fiber (PCTF) [1] is produced by combining phase change materials with conventional synthetic fibers [2], which are usually capable of intelligent temperature regulation [3,4]. Therefore, PCTF is widely used in aerospace, military [5], medical textiles [6,7] buildings and wearable systems [8]. PCTF has attracted much attention in the field of smart fibers and textiles.

At present, the well-known technologies for production of PCTF are electrospinning, wet spinning [9] and melt spinning [10]. Electrospinning cannot mass-produce PCTFs [11], and the use of organic solvents during the electrospinning process may cause environmental pollution. Wet spinning is a successful technology for mass production of PCTF, and its typical product is outlast^®^ fiber. Outlast fiber contains thermosensitive phase change materials in the form of microcapsules of 2–30 microns [12], which are implanted into viscose or acrylic fiber. Therefore, the thermal enthalpy (about 10 J/g) and the strength of outlast fiber is relatively low. Compared with the above technologies, melt spinning has the advantages of being solvent-free, environmentally friendly, a short process and high productivity [13].

However, most organic phase change materials have low thermal stability [14,15,16,17,18,19,20], and decompose below the temperature for melt spinning (above 220 °C). Therefore, how to improve the thermal stability of phase change materials to meet the requirement for melt spinning has become the key issue [21,22,23]. At the same time, suitable thermal enthalpy and the mechanical properties of PCTF were also required for practical applications in textiles. Zhang [24] developed a composite phase change material by combining paraffin wax with styrene-b-(ethylene-co-butylene-)-B-styrene triblock copolymer (SEBS) to improve its thermal stability. The results showed that the addition of SEBS did not affect the thermal properties of paraffin wax, and that the PCM could maintain good thermal stability below 200 °C with the increase of SEBS content. However, it still did not meet the requirement of melt spinning. Zhang and Wang [25] developed paraffin/CPCS (cross-linked phase change structure) composites, which improved the decomposition temperature from 0 °C to 120 °C. However, this was still far below the required temperature for melt spinning. In most studies, polyethylene [26], polyaniline [27], silicon dioxide [28] and expanded graphite [29] as substrates can improve the thermal stability of PCM. The combination of polyolefin and PCM has good rheological properties, and the thermal stability of PCM can be improved to meet the temperature requirements of melt spinning. Fang, et al., [30] introduced the preparation of microencapsulated composite phase change material with paraffin as phase change material and SiO_2_ as shell. The results showed that the thermal stability of paraffin is improved under the action of SiO_2_ shell. The internal paraffin decomposes in the temperature range of 200–350 °C, but the phase change temperature of the composite phase change material is 45–59 °C, which does not meet the temperature range of human thermal comfort. Therefore, it cannot be applied to the preparation of phase change fiber.

Cherif, et al., [31] described the preparation of thermoregulated fiber by melt spinning with a composite phase change material as the core layer and polypropylene as the skin layer, which has a thermoregulation range of about 44 °C and a latent heat value of up to 22 J/g. This was one of the most successful reports to date on the production of smart thermoregulated fiber by melt spinning, but the shortcomings are that its thermoregulation range is slightly above the comfort temperature range of the human body, and the tensile breaking strength (1.35 cN/dtex) is low, which makes it difficult for subsequent spinning and weaving. In addition, the composite fixed phase change material only meets the requirements for polypropylene melt spinning, and cannot be used for nylon, polyester and other products that require a higher spinning temperature (above 250 °C).

Polyethylene (PE) and polypropylene (PP) [26] are essentially chemically identical to the phase change material paraffin [32], so they can be used as a matrix for blending with PCM. However, there is a strong phase separation between components when paraffin is blended with polypropylene (PP) [33,34,35,36], which will reduce the thermal stability performance of paraffin to some extent. And the serrated structure of polyethylene crystals is more favorable to mutual compatibility with paraffin wax than the helical structure of polypropylene [32,37,38,39]. When polyethylene (PE) and paraffin wax (PW) are mixed, because it has good rheological properties, the polyethylene molecular chains are entangled at high temperature and have strong co-crystallization characteristics [40]. Therefore, polyethylene is the most suitable polymer to be blended with paraffin to improve the thermal stability of PCM [41,42,43] and to meet the temperature requirements of melt spinning [44]. The phase change temperature of the blended phase change material is 25–27 °C, which meets the temperature conditions for human thermal comfort. As the phase change material increases, although the latent heat of the fiber will increase, the PCM will experience mass loss and melt enthalpy loss rate increase at high temperature. The fiber strength will be reduced with subsequent use, so polyamide 6 (PA6), which is stronger as an outer layer, can improve the fiber strength and meet the actual production requirements for fiber strength.

Therefore, in this work, a composite phase change material was developed, by implanting paraffin into polyethylene to improve its thermal stability, and then a sheath-core structure PCTF was produced by bicomponent melt-spinning. The sheath material was polyamide 6 (PA6), and the core material was polyethylene (PE)/paraffin (PW). The PCTF was characterized regarding its structure, thermal stability, thermal storage properties and mechanical properties.

## 2. Materials and Methods

Polyamide 6 (PA6, ρ = 1.14 g/cm^3^, melting point about 240 °C) provided by Shanghai Bang Plastics New Materials Co., Ltd.; polyethylene (PE, melt index = 20 g/10 min, ρ = 0.91 g/cm^3^) provided by Beijing Kunpeng Chemical Co., Ltd.; paraffin (PW, ρ = 0.9 g/cm^3^, melting point = 58–60 °C) provided by Nantong Runfeng Petrochemical Co., Ltd.

### 2.1. Preparation of Phase Change Thermoregulated Fiber

Polyethylene/paraffin chips were made from polyethylene and paraffin at a mass ratio of 70:30 and then extruded through a TSE-30A twin-screw extruder at 150 r/min. By using a bicomponent melt spinning technology, a sheath-core structured fiber with polyamide 6 (PA6) as the sheath layer and polyethylene (PE)/paraffin wax (PW) as the core layer was produced. The sheath-to-core mass ratio was 6:4, the spinning spinneret contained 24 orifices with an aperture of 0.32 mm, the extrusion temperature was 210–225 °C for the sheath layer and 160–220 °C for the core layer, the spinning speed was 420 m/min, and the winding speed was 700 m/min. The schematic diagram of phase change thermoregulated fiber production is shown in Figure 1.

### 2.2. Testing and Characterization

Morphological observation: The surface and cross-sectional morphologies of the PCTF were observed by a JSM-5610LV SEM (JEOL, Tokyo, Japan) and a Phenom Pro benchtop SEM (Feiner, the Netherlands). The SEM detector was the secondary electron detector, and the accelerating voltage was 5 kV. The PCTF was fixed to the surface of the metal carrier, utilizing conductive adhesive, and the test surface was then sprayed with gold and tested.

Thermal decomposition performance test: in accordance with the GB/T 27761-2011 “Standard test method of mass loss and residue measurement validation of thermogravimetric analyzers” [45], the thermal decomposition performance of the PCTF was evaluated by a TA-Q500 thermal weight analyzer (American Parkin-Elmer company, Waltham, MA, USA). The PCTF was in the environment of N_2_, and the test temperature ranged from 0 °C–650 °C, with a temperature rise rate of 10 °C/min.

Chemical structure test: the functional groups and chain structure of the PCTF were tested by a VERTEX 70 FTIR spectrometer (Bruker Spectroscopy Instruments, Germany) in accordance with GB/T 6040-2019 “General Rules for Infrared Analysis” [46]; the test was carried out using the potassium bromide press method with a comprehensive range of 500–4000 cm^−1^.

Crystalline structure test: the crystallinity of the PCTF was tested by an ARL-XTRAX type radiographic powder diffractometer (Bruker AXS Ltd., Karlsruhe, Germany); the scanning range was 5–40°, and the scanning speed was 5(°)/min. The crystallinity of the PCTF was calculated according to Equation (1):
(1)$$W_{c.x}=I_{c}/\left(I_{c}+K_{x}I_{a}\right)$$


*W_c.x_* represents crystallinity; *I_c_* represents the integrated area of the diffraction peak in the crystalline state; *I_a_* represents the integrated area of the diffraction peak in the amorphous state; *K_x_* represents the total calibration factor.

Fiber enthalpy test: the Q2000 Differential Scanning Calorimeter (DSC) (TA Instruments, New Castle, DE, USA) was used to analyze the thermal energy storage capacity and stability of the PCTF. The testing temperature was 0–70 °C.

Mechanical properties test: the mechanical properties of the PCTF were evaluated by INSTRON 3369S3164 (INSTRON, USA). The mechanical properties of the PCTF were tested at different temperatures (−25 °C, −10 °C, 25 °C, 50 °C and 80 °C) when clamped at a gauge of 6 cm, respectively, and the tensile speed was 500 mm/min. Each sample was tested five times, and the results were averaged.

## 3. Results and Discussion

### 3.1. Morphology of the PCTF

The surface and cross-sectional morphology of the PCTF are shown in Figure 2. The surface of the PCTF was smooth, uniform and without impurities. There was a clear interface between the sheath layer and core layer, demonstrating a clear sheath-core structure. And the core materials were uniformly wrapped by the sheath layer.

### 3.2. Thermogravimetric Analysis of the PCTF

The relationship between the mass and the temperature of the PCTF is given in Figure 3. The TG curve of the PCTF had two stages. In the process of its temperature rise, the PCTF transformed from solid phase to liquid phase at 22–31 °C, and this process did not change the fiber mass fraction. The initial mass loss occurred from 230–350 °C. Because PE is a polymer material, it will decompose at more than 400 °C, so the first stage was the decomposition process of PW. PW accounted for 13.83% of the PCTF, which was slightly higher than the theoretical value. This indicates that the addition of paraffin and polyethylene effectively inhibits the decomposition of paraffin, improves the thermal stability of phase change material paraffin, and meets the temperature requirements for melt spinning. The second stage of weight loss occurred from 410–530 °C, which was a mixed decomposition process of PA6 and PE. In the thermal decomposition process, the cleavage of amide bonds and the cleavage of adjacent chemical bonds were the main products [47]. In the high temperature mixed decomposition process, the main products were CO, CO_2_, H_2_O and other derivatives [48]. Therefore, the fiber quality loss began to decline sharply. The results show that the thermal decomposition temperature of phase change material paraffin can be increased when paraffin is mixed with polyethylene. This indicates that the preparation of PCTF by melt spinning is promising.

### 3.3. Infrared Spectra of the PCTF

The IR spectrum of the PCTF is given in Figure 4. The peaks at 2952 cm^−1^, 2914 cm^−1^ and 2844 cm^−1^ were characteristic peaks for polyethylene (PE)/paraffin wax (PW). The –CH_3_–C–H asymmetric stretching vibration absorption peak was at 2952 cm^−1^. The stretching vibration absorption peak of –CH_2_–C–H was at 2914 cm^−1^. The –CH_3_–C–H symmetric stretching vibration absorption peak was at 2844 cm^−1^. The characteristic peak of polyamide 6 (PA6) was at 1737 cm^−1^, 1456 cm^−1^, 1373 cm^−1^ and 1103 cm^−1^. The stretching vibration absorption peak of –COOH–C=O was at 1737 cm^−1^. The in-plane bending vibration absorption peak of –CONH–C–H was at 1456 cm^−1^. The absorption peak of–OH–O–H in-plane bending vibration was at 1373 cm^−1^. The stretching vibration absorption peak of –NH_2_–C–H was at 1103 cm^−1^. These absorption peaks were characteristic of polyamide 6 (PA6) and the core material of polyethylene (PE)/paraffin (PW). They indicate that during the melt spinning process, the core material polyethylene (PE) and paraffin (PW) had no chemical reaction.

### 3.4. XRD Analysis of the PCTF

Figure 5 shows the X-ray diffraction curve of the PCTF at 2θ between 10°–30°. There were five obvious diffraction peaks at 2θ of 14.09°, 16.96°, 18.51°, 21.57° and 24.02°. Diffraction peaks at 2θ = 14.09° and 16.96° are the characteristic diffraction peak crystallographic planes of PE at (101) and (110). Diffraction peaks at 2θ = 21.57° and 24.02° are the characteristic diffraction peak crystallographic planes of PW at (200) and (201). A diffraction peak at 2θ = 18.51° is the characteristic diffraction peak of PA(6) at (111) [49]. Since each diffraction peak was wide, the crystal particles inside the fiber were small. The crystallinity of the PCTF was calculated by Formula (1) to be 63.28%, which indicated that there were many crystalline regions in the PCTF, and that the macromolecular chain structure was relatively neat and rigid.

### 3.5. DSC Analysis of the PCTF

The DSC curves of the PCTF are given in Figure 6. In the heating process, the solid-solid phase transformation of the phase change material paraffin (PW) occurred first, and then the solid-liquid phase transformation occurred. The cooling process was the reverse phase transformation of the heating process. The small peaks in the curve were due to the internal structure of the paraffin wax (PW) changing as the temperature rose; the internal lattice was changed to a flexible rotating body, and a small amount of heat was absorbed and released. This proved that the paraffin wax (PW) was only physically bonded to polyethylene (PE), and its crystalline pattern was not changed. The crystallization temperature was 24 °C, and the melting temperature was 29 °C.

As shown in Figure 6a, when the PCTF was heated, the phase change occurred at a temperature of 27.38–31.85 °C, forming a melting peak. Correspondingly, the enthalpy of melting (∆Hm) was 20.11 J/g. Similarly, the PCTF formed a crystallization peak at a temperature of 22.19–25.83 °C, corresponding to an enthalpy of crystallization (∆Hc) of 14.78 J/g. The results indicate that PCTF has good temperature regulation range and latent heat, which can be used for temperature regulation textiles.

### 3.6. Mechanical Properties of the PCTF

Figure 7 and Figure 8 show the stress-strain curves of the fibers at different clamping lengths at 25 °C. It can be seen from the images that the mechanical properties of the fibers followed approximately the same trend. When the clamping distance was 25 cm, and the fiber elongation was about 1800%, the fiber strength reached the maximum. When the clamping distance was 6 cm, and the fiber elongation was about 150%, the fiber strength reached the maximum. Due to the different clamping distances, the Young’s modulus of the fiber was also different. The modulus at 6 cm was 18.15 MPa, while the modulus at 25 cm was 15.14 MPa. The modulus of fiber with 6 cm was slightly larger than that with 25 cm. Due to the higher crystallinity of the fiber, the holding distance of the fiber was shorter, the amorphous region in the fiber was less, and the crystalline region was more, so the fiber modulus was larger. In the same stress range, the strain shown in Figure 7 was large, which was caused by the long clamping distance of the fiber.

The average stress-strain curves of the PCTF at different temperatures are given in Figure 9; all the PCTF demonstrated the typical stress-strain curve of conventional fiber, and had similar trends.

The PCTF was at its maximum stress at 0 °C, which was 343.59 MPa. However, the stress of the PCTF decreased to 296.77 MPa when the testing temperature was 80 °C, which decreased by 13.62%. The stress of the PCTF decreased to 284.34 MPa when the testing temperature was −25 °C, which decreased by 17.24% compared with the value at 0 °C. With the increase of testing temperature, the elongation of the PCTF improved to some extent. The maximum elongation reached 230% when the PCTF was at 80 °C. However, the maximum elongation decreased to 142% at −25 °C. These phenomena could be due to the phase change and to different forms at different temperatures, which influenced the mechanical properties of the PCTF.

From Table 1, it can be seen that the Young’s modulus of the PCTF showed a decrease with the temperature increase from 0 °C to 80 °C, which indicates that the rigidity of the PCTF was gradually decreasing, and that the ability of deformation was improved under external force. The modulus of the PCTF was 14.95 MPa at 0 °C, and was 12.65 Mpa at 80 °C, which decreased by 15.38%. The Young’s modulus of the PCTF increased from 0 °C to −25 °C; this was because the phase change materials underwent phase transition and heat absorption, and the crystallinity of the fibers increased with the decrease of temperature. The PCTF showed good mechanical properties for practical application in textiles.

From Table 1, it can be seen that there are obvious differences in the strength of the fiber at different temperatures of the PCTF. Figure 10 showed the strength and elongation of the PCTF at different temperatures. When the fiber was heated from −25 to 0 °C, the tenacity and elongation of the fiber increased to a certain extent, and the tenacity of the PCTF was at its greatest at 0 °C; as the temperature increased to 80 °C, the core layer of the fiber underwent phase change, which changed the internal structure of the fiber, so that although the fiber tenacity decreased, the change of tenacity tended to be stable and fluctuated less. Overall, the fiber had good tenacity and elongation at a room temperature of 25 °C, which meets the requirements of actual fiber production.

The Fracture work of the fiber at 0 °C was the largest (as shown in Figure 11), which was consistent with the maximum strength of the fiber at 0 °C in Figure 11, and the breaking energy of the PCTF at the rest of the temperatures tended to be consistent, which showed that the PCTF had the greatest strength and the strongest fatigue resistance at 0 °C. The strength of the PCTF tended to be consistent throughout the rest of the temperature range but was all slightly lower than the strength of the PCTF at 0 °C.

## 4. Conclusions

In this work, a composite chip was successfully prepared by combining polyethylene and paraffin wax, and the thermal stability of paraffin was improved. Thereafter, PCTF with sheath core structure was successfully produced by bicomponent melt spinning. The sheath layer was made from polyamide (PA6), and the core material was the blend of polyethylene (PE) and paraffin wax (PW).

The PCTF had a smooth surface and a clear structure of sheath-core layer. In the range of 20 °C−30 °C, the PCTF had obvious phase transition, and the enthalpy was 14.78–20.11 J/g. The mechanical properties of the PCTF fluctuated to some extent, and the tenacity of the PCTF tended to be stable with the increase of temperature. The PCTF reached its highest tenacity at 0 °C, with a tenacity of 343.59 MPa, and a tenacity of 254.54 MPa at 25 °C. The PCTF had good mechanical properties at all the different temperatures, and therefore fully met the production requirements for smart textiles. To further improve the thermal regulating performance of the PCTF, the future focus should be on changing supporting materials or adding nanoparticles to phase change materials for improving the enthalpy and thermal conductivity.

## Figures and Tables

**Figure 1 polymers-14-01895-f001:**
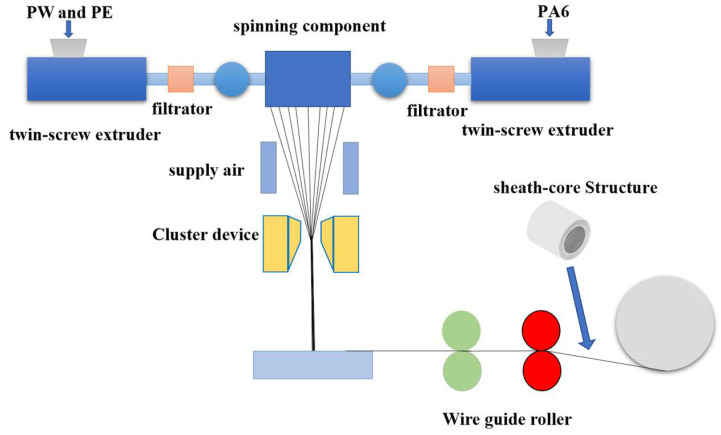
Schematic diagram of phase change thermoregulated fiber preparation.

**Figure 2 polymers-14-01895-f002:**
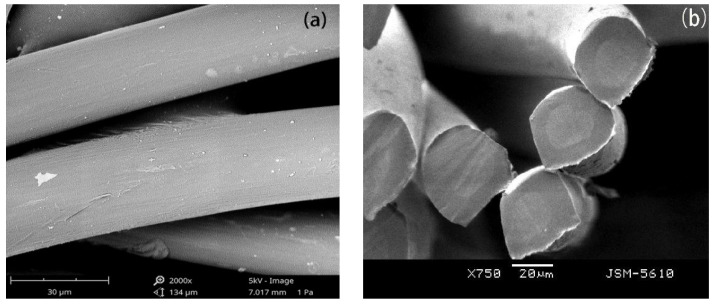
Surface and cross-sectional morphology of the PCTF (**a**) surface view (**b**) cross-sectional view.

**Figure 3 polymers-14-01895-f003:**
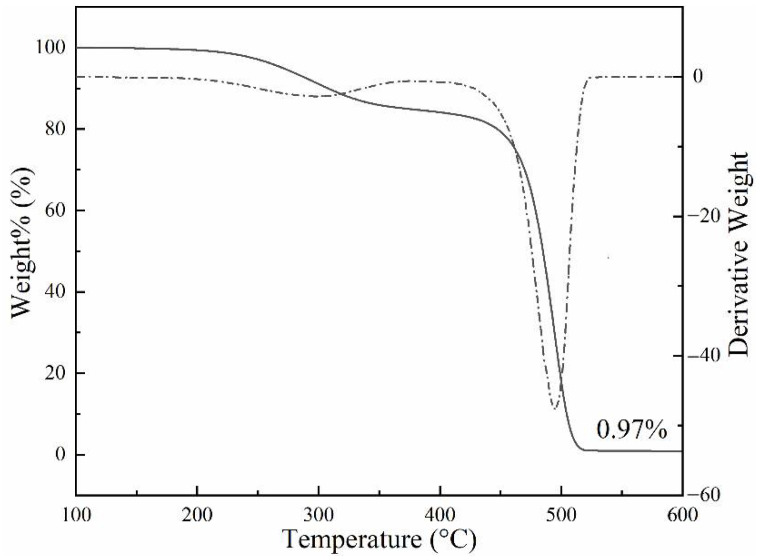
TG-DTG curves of the phase change thermoregulated fiber.

**Figure 4 polymers-14-01895-f004:**
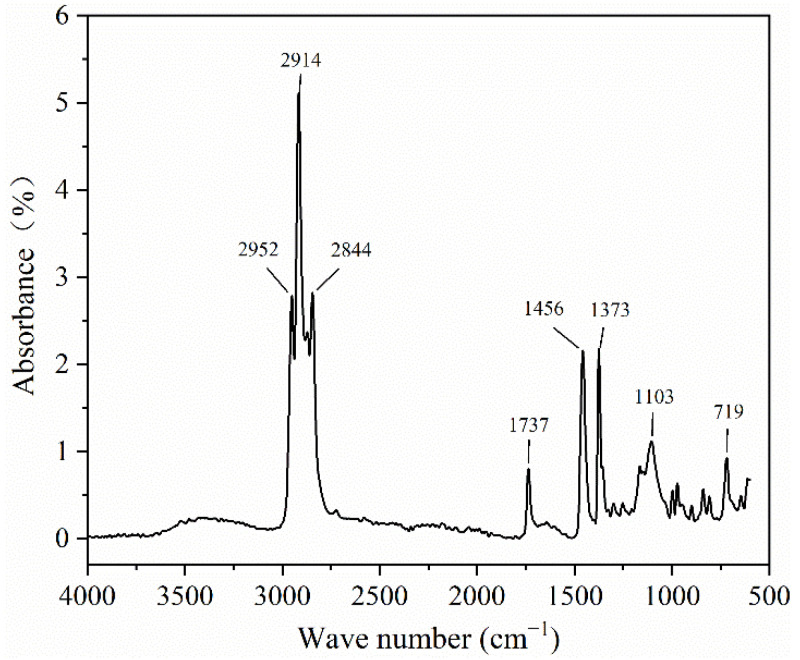
The infrared spectrum of the PCTF.

**Figure 5 polymers-14-01895-f005:**
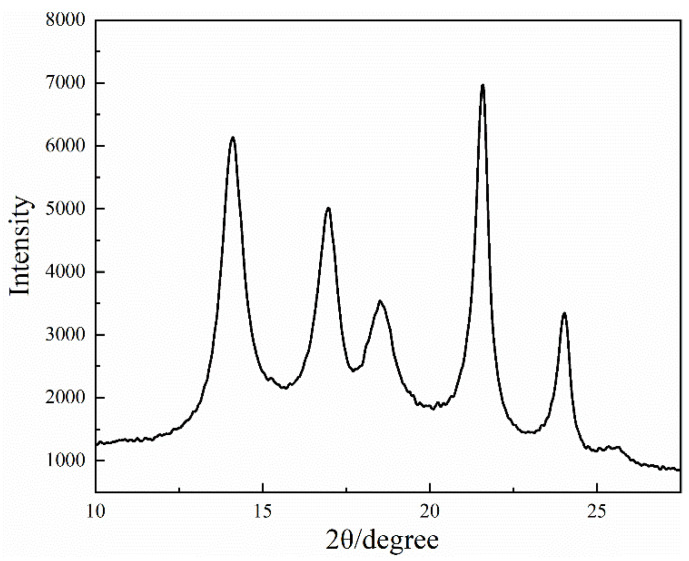
XRD curve of the PCTF.

**Figure 6 polymers-14-01895-f006:**
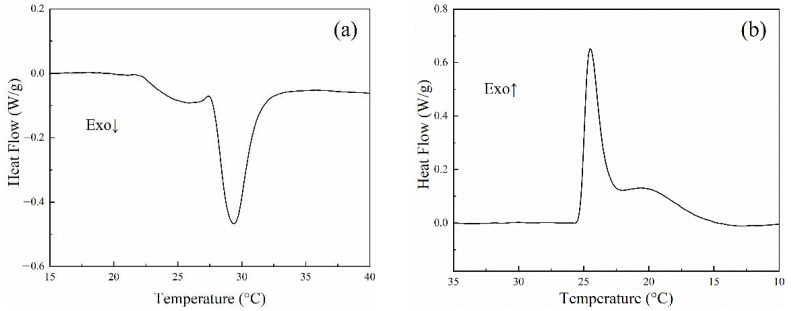
DSC curves for phase change thermoregulated fibers (**a**) heating curve (**b**) cooling curve.

**Figure 7 polymers-14-01895-f007:**
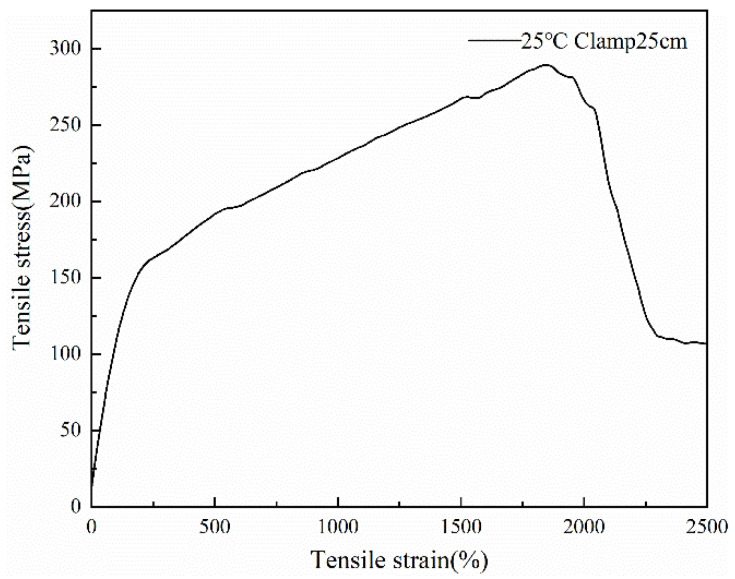
The stress-strain curve at 25 cm clamped at 25 °C.

**Figure 8 polymers-14-01895-f008:**
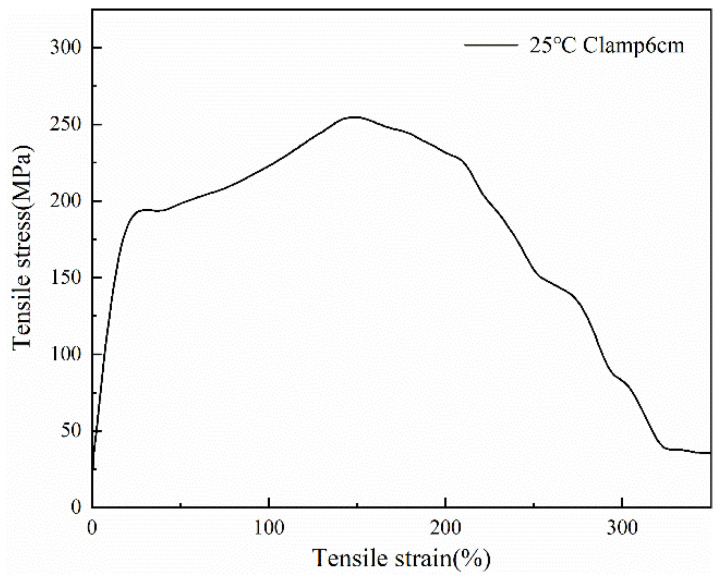
The stress-strain curve at 6 cm clamped at 25 °C.

**Figure 9 polymers-14-01895-f009:**
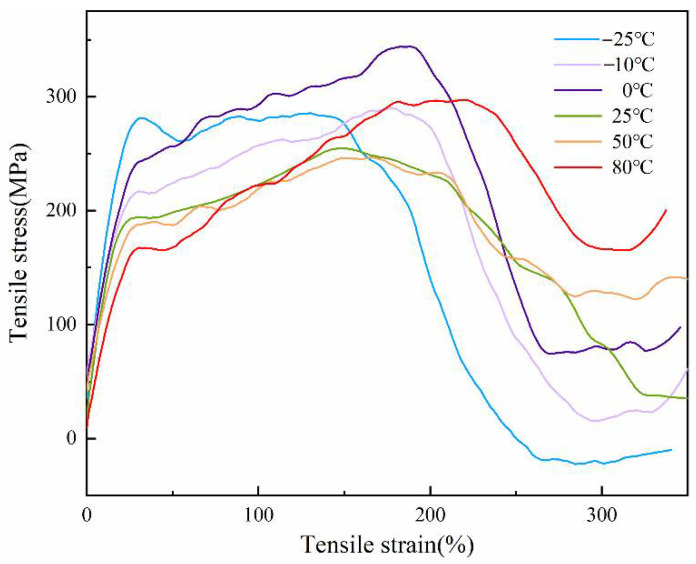
Stress-strain curves of the PCTF at different temperatures.

**Figure 10 polymers-14-01895-f010:**
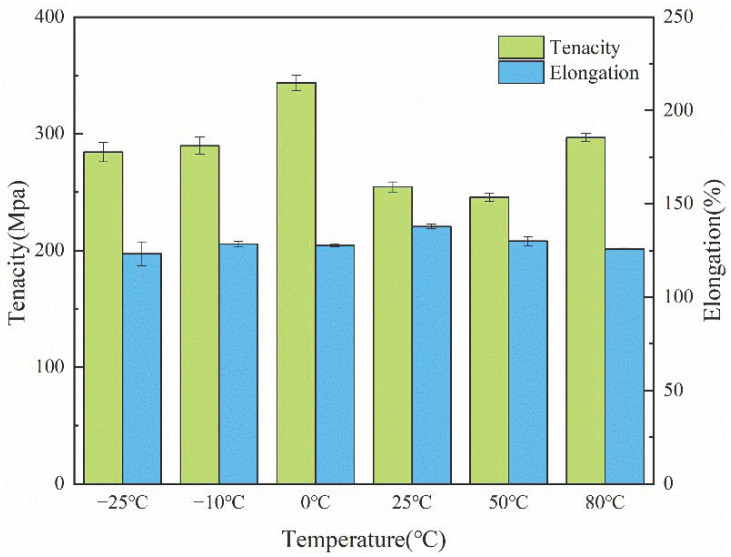
Tenacity and elongation of the PCTF at different temperatures.

**Figure 11 polymers-14-01895-f011:**
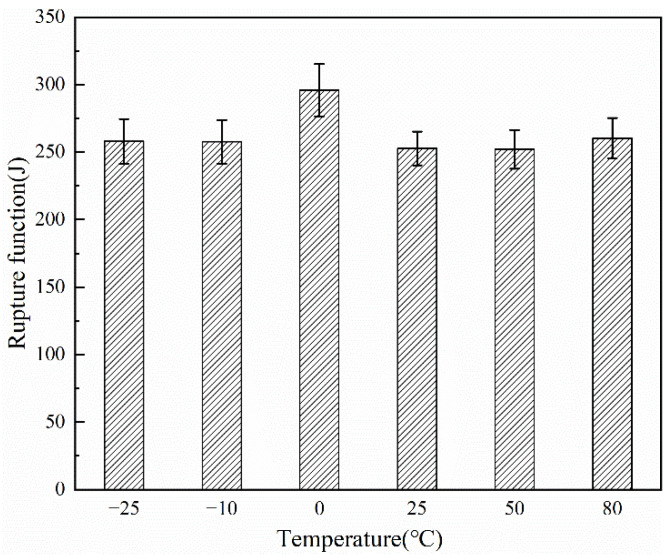
Fracture work of the PCTF at different temperatures.

**Table 1 polymers-14-01895-t001:** Mechanical properties of the PCTF at different temperatures.

**Temperature** **(°C)**	**Young’s Modulus (MPa)**	**Tenacity** **(MPa)**	**Elongation** **(%)**	**Fracture Work** **(J)**
−25	20.57 ± 0.61	284.34 ± 8.47	197.13 ± 10.29	258.02 ± 16.51
−10	13.74 ± 0.45	289.73 ± 7.48	205.49 ± 2.51	257.61 ± 16.05
0	14.95 ± 0.47	343.59 ± 6.63	204.52 ± 1.21	295.90 ± 19.45
25	18.15 ± 0.42	254.54 ± 4.31	220.58 ± 2.08	252.71 ± 12.47
50	12.65 ± 0.32	245.63 ± 3.28	208.08 ± 3.78	252.15 ± 14.36
80	11.86 ± 0.37	296.77 ± 3.67	201.35 ± 0.56	260.20 ± 15.07

## Data Availability

Data is contained within the article.

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
