# Peer review of "Fabrication and Performance of Phase Change Thermoregulated Fiber from Bicomponent Melt Spinning"

_polymers, 2022, doi:10.3390/polym14091895_

Round 1

Reviewer 1 Report

This work deals with phase change thermoregulated fibers (PCTF) produced by melt spinning.

THE WHOLE WORK IS INTERESTING AND COULD HAVE POTENTIAL APPLICATIONS.

POINTS FOR IMPROVEMENT:

  1. Please, report some additional technical data for reagents used. For example in PE the product density could be added.
  2. What's the limitation of the proposed method?
  3. What's the main problem for industrial application?
  4. Propose ideas for future work.
  5. Please, make a brief patent literature

In my opinion this paper could be published after revision.

Reviewer 2 Report

This is an interesting work devoted to the actual topic of obtaining smart textiles.

The paper provides a brief overview and succinctly presents interesting results worthy of publication after correcting some minor flaws:

  1. Line 2. At the beginning of the manuscript, the authors and organizations should be indicated.
  2. Line 77 The PW abbreviation should be defined here because here it is mentioned for the first time
  3. Line 106 what is the "spinning speed" means?
  4. Lines 106-107 420 and 700 m/s Is that a typo? It is impossible to spin at that high speed
  5. Figure 1 what is a tensile strain device?
  6. Line 110 The detector type and the accelerating voltage must be given
  7. Lines 113 and 120 The references on GB/T 27761-2011 and GB/T 6040-2019 should be added.
  8. Line 162, etc. The wavenumbers should be rounded to significant values
  9. Figure 6 “Eco” on Fig.6a should be replaced on Exo, “Endo” should be also replaced on “Exo” with corresponding changing the arrow direction
  10. Figure 9. What distance between clamps is used? How reproducible are these results?
  11. Line 260 Polymers is not a textile journal. Modulus and fiber strength must be given in SI units (Pa).

Reviewer 3 Report

The manuscript entitled "Fabrication and performance of phase change thermoregulated fiber from bicomponent melt spinning" presents an interesting approach, but many drawbacks have to be solved.

  • The word for highest strength is TENACITY.
  • Editing is needed. A space should be introduced before the citations.
  • Please reformulate: "The structure, thermal stability, thermal storage properties, and mechanical properties of PCTF were characterized.", for instance: PCTF was characterized regarding its structure, thermal stability, thermal storage properties, and mechanical properties.
  • Line 147, please Correct The relationship............is given
  • Figure 3, does not correspond to the description text.
  • The IR spectrum, does not need peaks measured with such high precision.  2952.04 should be 2952 cm-1................
  • XRd needs a more detailed discussion.
  • DSC should be discussed together with TG-DTG.
  • Terms lack to be used as expected: modulus of elasticity or Young's modulus
  • References should be completed with many more recent reports.
